# Healthcare in England was affected by the COVID-19 pandemic across the pancreatic cancer pathway: A cohort study using OpenSAFELY-TPP

**Agnieszka Lemanska[1]\*, Colm Andrews[2], Louis Fisher[2], Seb Bacon[2], Adam E Frampton[1,3,4], Amir Mehrkar[2], Peter Inglesby[2], Simon Davy[2], Keith Roberts[5], Praveetha Patalay[6], Ben Goldacre[2], Brian MacKenna[2], The OpenSAFELY Collaborative, Alex J Walker[2]**

[1]Faculty of Health and Medical Sciences, University of Surrey, Guildford, United Kingdom; [2]Bennett Institute for Applied Data Science, Nuffield Department of Primary Care Health Sciences, University of Oxford, Oxford, United Kingdom; [3]HPB Surgical Unit, Royal Surrey NHS Foundation Trust, Guildford, United Kingdom; [4]Oncology Section, Surrey Cancer Research Institute, Department of Clinical and Experimental Medicine, University of Surrey, Guildford, United Kingdom; [5]Institute of Immunology and Immunotherapy, University of Birmingham, Birmingham, United Kingdom; [6]MRC Unit for Lifelong Health and Ageing and Centre for Longitudinal Studies, University College London, London, United Kingdom

**\*For correspondence:**
a.lemanska@surrey.ac.uk

**Group author details:**
The OpenSAFELY Collaborative
See page 14

## Abstract

**Background:** Healthcare across all sectors, in the UK and globally, was negatively affected by the COVID-19 pandemic. We analysed healthcare services delivered to people with pancreatic cancer from January 2015 to March 2023 to investigate the effect of the COVID-19 pandemic.

**Methods:** With the approval of NHS England, and drawing from a nationally representative Open-SAFELY-TPP dataset of 24 million patients (over 40% of the English population), we undertook a cohort study of people diagnosed with pancreatic cancer. We queried electronic healthcare records for information on the provision of healthcare services across the pancreatic cancer pathway. To estimate the effect of the COVID-19 pandemic, we predicted the rates of healthcare services if the pandemic had not happened. We used generalised linear models and the pre-pandemic data from January 2015 to February 2020 to predict rates in March 2020 to March 2023. The 95% confidence intervals of the predicted values were used to estimate the significance of the difference between the predicted and observed rates.

**Results:** The rate of pancreatic cancer and diabetes diagnoses in the cohort was not affected by the pandemic. There were 26,840 people diagnosed with pancreatic cancer from January 2015 to March 2023. The mean age at diagnosis was 72 (±11 SD), 48% of people were female, 95% were of White ethnicity, and 40% were diagnosed with diabetes. We found a reduction in surgical resections by 25–28% during the pandemic. In addition, 20%, 10%, and 4% fewer people received body mass index, glycated haemoglobin, and liver function tests, respectively, before they were diagnosed with pancreatic cancer. There was no impact of the pandemic on the number of people making contact with primary care, but the number of contacts increased on average by 1–2 per person amongst those who made contact. Reporting of jaundice decreased by 28%, but recovered within 12 months into the pandemic. Emergency department visits, hospital admissions, and deaths were not affected.

**Conclusions:** The pandemic affected healthcare in England across the pancreatic cancer pathway. Positive lessons could be learnt from the services that were resilient and those that recovered quickly. The reductions in healthcare experienced by people with cancer have the potential to lead to worse outcomes. Current efforts should focus on addressing the unmet needs of people with cancer.

**Funding:** This work was jointly funded by the Wellcome Trust (222097/Z/20/Z); MRC (MR/V015757/1, MC_PC-20059, MR/W016729/1); NIHR (NIHR135559, COV-LT2-0073), and Health Data Research UK (HDRUK2021.000, 2021.0157). This work was funded by Medical Research Council (MRC) grant reference MR/W021390/1 as part of the postdoctoral fellowship awarded to AL and undertaken at the Bennett Institute, University of Oxford. The views expressed are those of the authors and not necessarily those of the NIHR, NHS England, UK Health Security Agency (UKHSA), or the Department of Health and Social Care. Funders had no role in the study design, collection, analysis, and interpretation of data; in the writing of the report; and in the decision to submit the article for publication.

## Editor's evaluation

This study provides useful information on the impact of the pandemic on the quantity of healthcare delivered to patients with pancreatic cancer in England. The authors showed that there was no difference in the number of diagnoses of pancreatic cancer during the pandemic compared to the preceding 5-year period, but a reduction in surgical resections by nearly 25%. They reported no difference in deaths between the two periods. They show no differences in rates of diagnosis, but the clinical relevance is incomplete as they have not compared survival from cancer between those time periods.

## Introduction

Cancer services were already overstretched before the COVID-19 pandemic (**NHS, 2019**; **NHS 75 England, 2019**). With the widespread effect on healthcare, the pandemic further exacerbated the cancer-related healthcare crisis (**Greenwood and Swanton, 2021**; **Patt et al., 2020**; **Richards et al., 2020**; **Morris et al., 2021**; **McKay et al., 2021**; **Glasbey et al., 2021**; **Nepogodiev et al., 2022**; **Diamand et al., 2021**; **Sud et al., 2020**; **Earnshaw et al., 2020**; **Geh et al., 2022**; **Popovic et al., 2022**). During the pandemic, the resources, and the attention in healthcare systems globally, shifted towards preventing and managing COVID-19 (**Mercier et al., 2020**; **Williams et al., 2020**). Access to the non-COVID-19-related healthcare changed (**Núñez et al., 2021**; **Smolić et al., 2022**; **Vardhanabhuti and Ng, 2021**), waiting times increased (**Mazidimoradi et al., 2023**; **Cooke et al., 2022**), and cancer pathways including treatment standards were adapted (**Greenwood and Swanton, 2021**; **Patt et al., 2020**; **Richards et al., 2020**; **Morris et al., 2021**; **McKay et al., 2021**; **Glasbey et al., 2021**; **Nepogodiev et al., 2022**; **Diamand et al., 2021**). In addition, patients' healthcare-seeking behaviour changed as people adopted social distancing (limiting face-to-face contact) and shielding to protect themselves and healthcare systems from unprecedented pressures of the pandemic (**Robb et al., 2020**; **Smith et al., 2022**; **Hughes et al., 2020**). People were cautious and were actively taking measures to preserve healthcare and limit the spread of COVID-19 (**Quinn-Scoggins et al., 2021**).

People affected by cancer were particularly vulnerable to the changes brought by the pandemic (**Support, 2020**). This is because they rely on healthcare. In pancreatic cancer, the challenging diagnosis (due to non-specific symptoms) and rapid progression require an efficient system (**Zhang et al., 2018**). Weight loss, hyperglycaemia, diabetes, and bile duct obstruction often occur as complications of pancreatic cancer (**Mueller et al., 2019**). Therefore, timely assessments of body mass index (BMI), glycated haemoglobin (HbA1c), and liver function can support early diagnosis (**Lemanska et al., 2022**) and monitoring of the progression (**Sharma et al., 2018**). However, the negative effect of the COVID-19 pandemic resulted in delays and missed opportunities throughout the cancer pathway, which in turn affected patient outcomes including survival (**Morris et al., 2021**; **McKay et al., 2021**; **Glasbey et al., 2021**; **Nepogodiev et al., 2022**; **Sud et al., 2020**; **Earnshaw et al., 2020**; **Geh et al., 2022**; **Mazidimoradi et al., 2023**; **appg, 2021**).

To mitigate the negative effect of the COVID-19 pandemic and support patients and healthcare systems in recovery, it is important to provide the assessment of the scale of the impact. We therefore set out to investigate the effect of the pandemic on pancreatic cancer services in England. The objectives were to:

1. Access nationally representative data on healthcare services across the pancreatic cancer pathway. This was to investigate the effect of the COVID-19 pandemic on the range of services, from diagnostics to survivorship, as well as different healthcare settings including primary and secondary care.
2. Compare the quantity of healthcare that would be delivered if the pandemic had not happened (predicted based on the pre-pandemic trends) to that actually delivered (observed) during the pandemic. This was to assess the effect of the COVID-19 pandemic using the significance of the difference between the observed and predicted rates.
3. Access near real-time longitudinal data (up to March 2023) and analyse trends over time. This was to investigate patterns in the recovery of services from the effect of the pandemic.

## Methods
### Study design
This was a cohort study set in England, UK. We analysed electronic healthcare records (EHR) of adults diagnosed with pancreatic cancer between 1 January 2015 and 31 March 2023.

### Data source: OpenSAFELY-TPP dataset
We used the OpenSAFELY-TPP dataset comprising 24 million people currently registered with primary care practices that use TPP's SystmOne software (covering over 40% of England's population). This dataset was used for this project because of its unprecedented size, because it is nationally representative (*Andrews et al., 2022*), and because it enables access to primary care records linked to hospital records and mortality data. Linked pseudonymised EHRs included coded diagnoses, medications, and physiological parameters. No free text data were available.

Primary care records managed by TPP were linked to Secondary Uses Service (SUS) hospital procedures and admissions data, and to Office of National Statistics (ONS) death data through OpenSAFELY. OpenSAFELY is an analytics platform created by our team on behalf of NHS England to address urgent COVID-19 research questions. It provides a secure software interface allowing the analysis of pseudonymised primary care records of patients from England in near real time within the TPP's highly secure data centre, avoiding the need for large volumes of patient data to be transferred off-site. This, in addition to other technical and organisational controls, minimises any risk of patient re-identification. Further details can be found in the information governance section of this manuscript and on opensafely.org.

### Study population
The study population was derived from the 24 million people in the OpenSAFELY-TPP dataset. Participants were adults diagnosed with pancreatic cancer between 1 January 2015 and 31 March 2023. Data on healthcare services delivered to study participants were extracted and trends over time were analysed.

### Outcome measures
The information on new pancreatic cancer diagnosis (incidence) was extracted. This was defined as the first time that a clinical code for pancreatic cancer was entered in a primary care record. The age of participants at diagnosis, their gender and ethnicity were also extracted. To assess the effect of the pandemic on pancreatic cancer diagnosis, numbers of people diagnosed with pancreatic cancer were presented as monthly rates per 100,000 adults registered with TPP practices.

To analyse the effect of the COVID-19 pandemic on pancreatic cancer-related services, the episodes of care were extracted from 6 months before to 6 months after pancreatic cancer diagnosis (apart from the diabetes diagnosis which was extracted at any time). The date of pancreatic cancer diagnosis served as an index date for each participant. *Table 1* presents which healthcare services were included in the study, and the time windows for which they were assessed.

**Table 1.** Healthcare services (contacts, appointments, diagnosis, diagnostic tests, routine assessments) and associated time windows for which they were extracted.

Pancreatic cancer diagnosis was an index date. Counts of healthcare services were presented as either monthly rates of people who were diagnosed that month and received a healthcare service within the time window per 100 people diagnosed that month, or number of episodes received within the time window per person diagnosed each month (except for the number of primary care contacts which was analysed per person in contact).

| Healthcare service | Time window | |
|---|---|---|
| | Six months before pancreatic cancer diagnosis | Six months after pancreatic cancer diagnosis |
| **Primary care** | | |
| Diabetes diagnoses | Any time | |
| Contacts with primary care (these include all contact events, not only healthcare appointments) | ✓ | ✓ |
| BMI assessments | ✓ | ✓ |
| HbA1c assessments | ✓ | ✓ |
| Liver function assessments | ✓ | ✓ |
| Reporting of jaundice | ✓ | ✗ |
| Pancreatic enzyme supplementation | ✗ | ✓ |
| **Secondary care** | | |
| Abdominal imaging | ✓ | ✗ |
| Pancreatic cancer resection (surgery) | ✗ | ✓ |
| Emergency department visits | ✓ | ✓ |
| Hospital admissions | ✓ | ✓ |
| **National mortality register** | | |
| Death (any cause) | ✗ | ✓ |

Data on healthcare services were assessed as monthly rates of people who received a healthcare service (≥one episode) per 100 people diagnosed. Some services were assessed as monthly rates (numbers) of episodes per one person diagnosed (or for primary care contacts this was per person in contact).

Primary care data were extracted using the systematised nomenclature of medicine clinical terminology (SNOMED CT) system. Medications data, namely pancreatic enzyme supplements, were extracted using the list compiled based on the British National Formulary (BNF) and coded using NHS Dictionary of Medicines and Devices codes. Hospital procedures data were queried using the Office of Population Censuses and Surveys (OPCS-4) coding system.

## Study dates

The study period was from 1 January 2015 to 31 March 2023. In the UK, the pandemic-related restrictions started in March 2020 with the first national lockdown in England commencing on 26 March 2020, and the two consecutive lockdowns starting 5 November 2020 and 6 January 2021. From 8 March 2021, governments in the UK began a phased exit from the third and final lockdown. Therefore, to analyse the effect of the COVID-19 pandemic, three separate periods were adopted. The period before the pandemic was from 1 January 2015 to 29 February 2020. The lockdown period was from 1 March 2020 to 31 March 2021. The recovery period (the period of easing restrictions) was from 1 April 2021 to 31 March 2023. The recovery period was censored 6 months earlier, by 30 September 2022, for the outcomes that assessed healthcare service 6 months after pancreatic cancer diagnosis. This was to ensure the completeness of the 6-month follow-up data.

Primary care data were available for the whole study period. However, secondary care data were only available from January 2017 onwards. Emergency department visits as well as ONS mortality data were available from January 2019 onwards.

### Statistical analysis

Counts of patients and healthcare services were rounded to the nearest 5 to comply with the rules for preventing statistical disclosure. The observed monthly rates were visualised between 1 January 2015 (or as available) and 31 March 2023 (or 30 September 2022 for the outcomes that assessed healthcare within 6 months after pancreatic cancer diagnosis).

Data from before the pandemic were used to predict monthly rates of healthcare services that would be expected during the lockdown and recovery periods if the pandemic had not happened. Generalised linear models (GLM) were used to model monthly rates. A separate model was fitted for each healthcare service. An interrupted time series approach was used to predict the expected rates in the lockdown and recovery periods. To account for seasonality in data, calendar months were fitted as a categorical variable. To allow for change in healthcare services over time, the time was fitted as a continuous variable. Two dummy variables for the two COVID-19 periods were included to allow trends and slopes to vary in these periods separately.

The differences between the observed and predicted rates were calculated and presented as the percentage change from the predicted. The 95% confidence intervals (CIs) of the predicted values were used to estimate the significance of the difference between the predicted and observed values (to estimate the effect of the COVID-19 pandemic). The average values across periods (rather than any specific points in time) were used to estimate the overall effect in each period.

### Software and reproducibility

Data management was performed in SQL and Python version 3.8. Statistical analyses were performed in R version 4.0.2 using packages MASS for GLM and ggplot2 for data visualisation. The REporting of studies Conducted using Observational Routinely-collected health Data (RECORD) guidelines were followed (*Benchimol et al., 2015*). Software for data analysis and code lists used to define outcome measures are available from https://github.com/opensafely/Pancreatic_cancer (copy archived at *Lemanska, 2023*).

## Results

### Study population and participants

In total, there were 26,840 people diagnosed with pancreatic cancer in the study period (study participants). On average, there were 267 (±24 SD) pancreatic cancer diagnosis each month. The mean age at pancreatic cancer diagnosis was 72 (±11 SD). 12,965 (48%) participants were females and 13,875 (52%) were males. 18,760 participants were of White ethnicity (95% for which ethnicity data were recorded). Ethnicity data were missing for 7040 (26%) participants. 10,785 (40%) of people with pancreatic cancer received diagnosis of diabetes at some point before or after their cancer diagnosis.

### The effect of COVID-19 on pancreatic cancer and diabetes diagnosis

We did not observe an effect of the COVID-19 pandemic on the number of people recorded as diagnosed with pancreatic cancer (*Figure 1A*). For every 100,000 registered people, there was 1 pancreatic cancer diagnosis a month in the lockdown and in the recovery period. This equated to the predicted rate of 1 (95% CI: 1–2) in both periods of the pandemic (*Table 2*). We also did not observe an effect on diabetes diagnosis in this cohort (*Figure 1B*). In both periods of the pandemic, 41% of people received diabetes diagnosis. The predicted rates per 100 diagnosed people were 41 (95% CI: 38–44) in the lockdown period and 42 (95% CI: 39–45) in the recovery period.

### Primary care contacts

The number of people with pancreatic cancer who contacted primary care before and after their diagnosis, increased over time, from 70% and 80% (respectively) in 2015 to over 90% by 2022 (*Figure 2A, B*). This trend was not affected by the pandemic (*Table 2*). We observed that 91% (predicted 92% [95% CI: 90–94]) of people diagnosed in the lockdown period, contacted (≥1 contact) primary care within

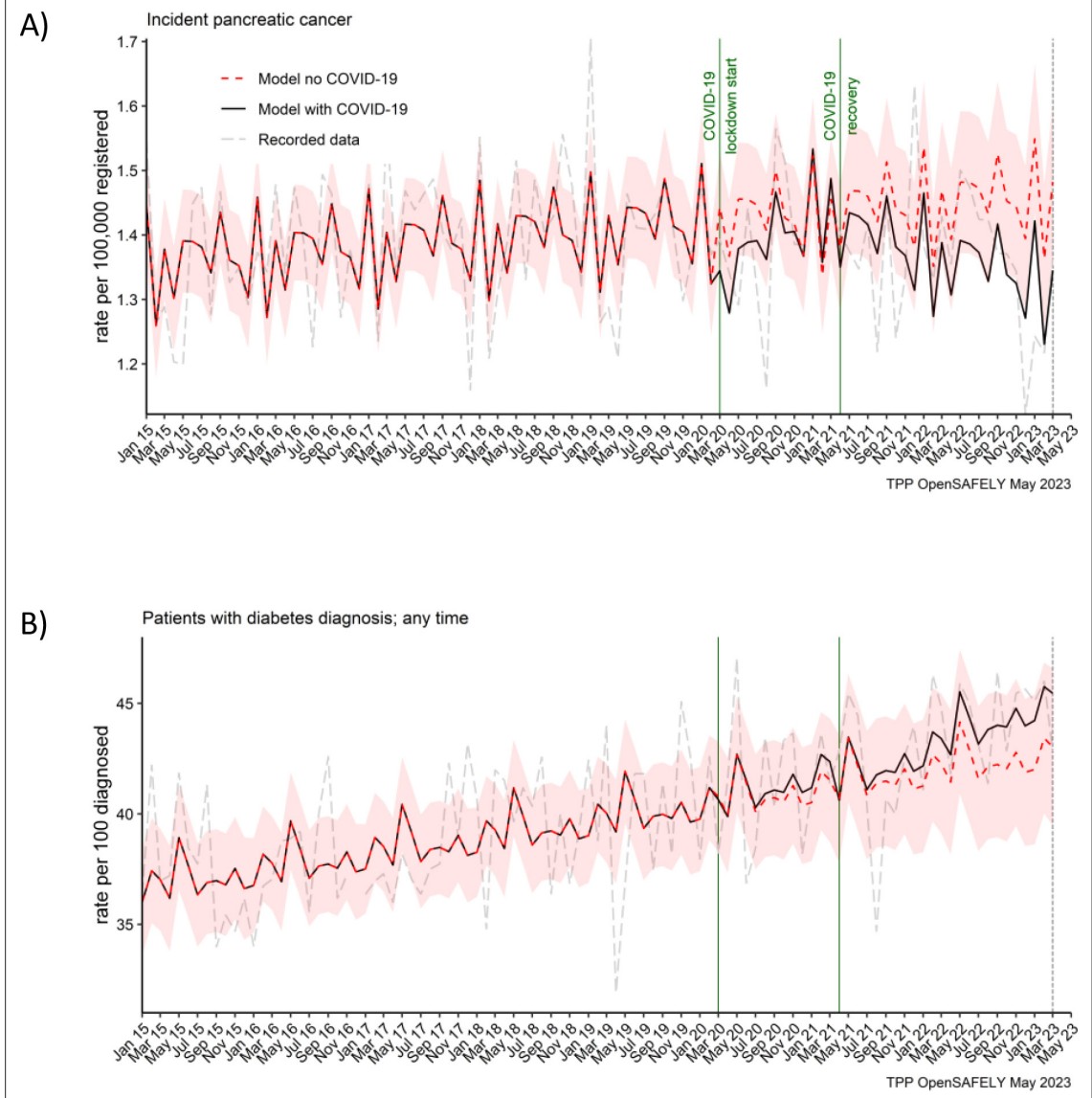

**Figure 1.** Pancreatic cancer diagnosis and diabetes diagnosis in people with pancreatic cancer were not affected by the pandemic. Observed and predicted monthly rates (as if the COVID-19 pandemic had not happened) for (**A**) pancreatic cancer diagnosis per 100,000 registered patients, and (**B**) diabetes diagnosis per 100 people with pancreatic cancer. Generalised linear models were used to predict monthly rates with the 95% confidence intervals to estimate the effect of the pandemic.

6 months before they were diagnosed, and 88% (predicted 87% [95% CI: 84–89]) contacted primary care within 6 months after diagnosis. In the recovery period, this was 95% (predicted 97% [95% CI: 95–99]) and 91% (predicted 91% [95% CI: 89–94]), respectively.

However, there was a difference in the number of primary care contacts recorded per person. People diagnosed in the lockdown period had on average 1 contact more than predicted (12 vs 11 [95% CI: 10–12]) within 6 months before the diagnosis and 2 contacts more than predicted (13 vs 11 [95% CI: 10–11]) within 6 months after they were diagnosed. This was similar for people diagnosed in the recovery period. For these people, there were 2 more contacts observed than predicted (13 vs 11 [95% CI: 10–12]) before the diagnosis and 2 (12 vs 11 [95% CI: 10–12]) after the diagnosis (*Table 2*). Any discrepancies in sums are due to rounding. In addition, it is important to note that contacts with primary care included all the contact events (all reasons and purposes), not only health-care appointments.

**Table 2.** Statistical significance of the differences between the observed and predicted monthly rates of healthcare services with 95% confidence intervals.

The pandemic values were predicted based on the pre-pandemic period from 1 January 2015 to 29 February 2020. The lockdown period was from 1 March 2020 to 31 March 2021. The recovery period (the period of easing restrictions) was from 1 April 2021 to 31 March 2023 (or to 30 September 2022 for healthcare services that were evaluated 6 months after diagnosis). The values are the average monthly rates over the period. Unless otherwise specified, the rates are per 100 people diagnosed with pancreatic cancer. *Indicates statistical significance with 95% confidence levels.

| | Lockdown period: 1 March 2020 to 31 March 2021 (13 months) | | | Recovery period: 1 April 2021 to 31 March 2023 (24 months) or 30 September 2022 (18 months for services evaluated 6 months after diagnosis) | | |
|---|---|---|---|---|---|---|
| | Predicted rates (95% CI) | Observed rates | Difference | Predicted rates (95% CI) | Observed rates | Difference |
| Pancreatic cancer diagnosis (rate per 100,000 people registered) | 1 (1–2) | 1 | 0 (2%) | 1 (1–2) | 1 | 0 (6%) |
| Diabetes diagnosis any time before or after pancreatic cancer diagnosis | 41 (38–44) | 41 | 0 (1%) | 42 (39–45) | 43 | 1 (3%) |
| People with ≥1 primary care contacts within 6 months before pancreatic cancer diagnosis | 92 (90–94) | 91 | −1 (1%) | 97 (95–99) | 95 | −2 (3%)* |
| People with ≥1 primary care contacts within 6 months after pancreatic cancer diagnosis | 87 (84–89) | 88 | 2 (2%) | 91 (89–94) | 91 | 0 (0%) |
| Number of primary care contacts within 6 months before pancreatic cancer diagnosis (per person in contact) | 11 (10–12) | 12 | 1 (8%)* | 11 (10–12) | 13 | 2 (15%) |
| Number of primary care contacts within 6 months after pancreatic cancer diagnosis (per person in contact) | 11 (10–11) | 13 | 2 (18%)* | 11 (10–12) | 12 | 2 (14%)* |
| People with ≥1 BMI assessments within 6 months before pancreatic cancer diagnosis | 55 (52–59) | 44 | −11 (20%)* | 57 (53–61) | 49 | −8 (14%)* |
| People with ≥1 BMI assessments within 6 months after pancreatic cancer diagnosis | 31 (28–34) | 25 | −6 (18%)* | 33 (30–36) | 31 | −2 (7%) |
| People with ≥1 HbA1c assessments within 6 months before pancreatic cancer diagnosis | 60 (57–64) | 55 | −6 (10%)* | 66 (62–70) | 61 | −5 (7%)* |
| People with ≥1 HbA1c assessments within 6 months after pancreatic cancer diagnosis | 19 (16–21) | 16 | −3 (13%) | 20 (17–22) | 18 | −1 (7%) |
| People with ≥1 liver function assessments within 6 months before pancreatic cancer diagnosis | 80 (77–82) | 76 | −3 (4%)* | 80 (77–83) | 78 | −2 (3%) |
| People with ≥1 liver function assessments within 6 months after pancreatic cancer diagnosis | 34 (31–36) | 33 | −1 (2%) | 33 (30–36) | 33 | 1 (2%) |
| People reporting jaundice ≥1 times within 6 months before pancreatic cancer diagnosis | 10 (9–11) | 7 | −3 (28%)* | 10 (9–12) | 9 | −1 (11%) |
| People receiving abdominal scan ≥1 times within 6 months before pancreatic cancer diagnosis | 30 (27–33) | 29 | −2 (5%) | 30 (26–34) | 29 | 0 (2%) |
| People receiving ≥1 prescriptions for enzyme supplements within 6 months after pancreatic cancer diagnosis | 55 (52–58) | 54 | −2 (3%) | 60 (56–63) | 57 | −3 (5%) |
| People receiving pancreatic cancer resection within 6 months after pancreatic cancer diagnosis | 8 (6–10) | 6 | −2 (25%)* | 9 (7–11) | 6 | −2 (28%)* |
| Number of emergency department visits within 6 months before pancreatic cancer diagnosis (per person diagnosed) | 1 (1–1) | 1 | 0 (9%) | 1 (0–1) | 1 | 0 (11%) |
| Number of emergency department visits within 6 months after pancreatic cancer diagnosis (per person diagnosed) | 1 (0–1) | 1 | 0 (1%) | 1 (0–1) | 1 | 0 (5%) |
| Number of hospital admissions within 6 months before pancreatic cancer diagnosis (per person diagnosed) | 2 (2–2) | 2 | 0 (7%) | 2 (1–2) | 2 | 0 (3%) |
| Number of hospital admissions within 6 months after pancreatic cancer diagnosis (per person diagnosed) | 4 (4–5) | 4 | 0 (2%) | 4 (3–5) | 4 | 0 (8%) |
| People who died within 6 months after pancreatic cancer diagnosis | 61 (53–69) | 56 | −5 (8%) | 68 (53–82) | 56 | −12 (17%) |

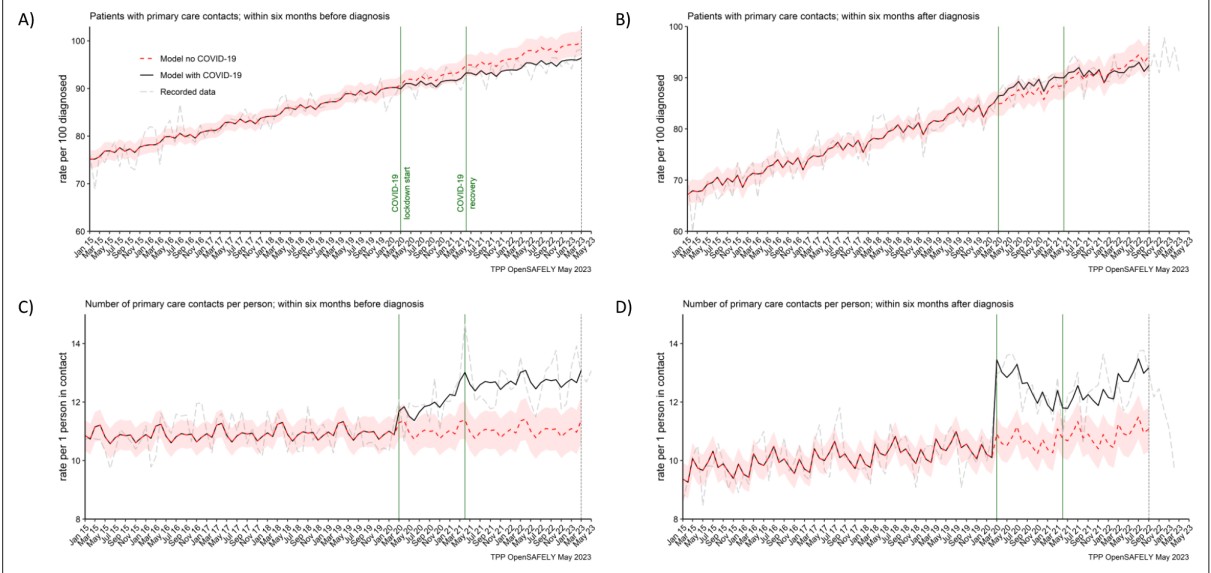

**Figure 2.** The number of people contacting primary care was not affected, but the number of contacts per person increased during the pandemic. The observed and predicted (as if the COVID-19 pandemic had not happened) monthly rates of (**A**) people in contact with primary care before pancreatic cancer diagnosis, and (**B**) people in contact with primary care after pancreatic cancer diagnosis. Rates are per 100 people diagnosed with pancreatic cancer. Figures (**C**) and (**D**) are monthly rates of primary care contacts per person before and after pancreatic cancer diagnosis. Contacts with primary care include all contact events (all reasons and purposes), not only healthcare appointments. Generalised linear models were used to predict monthly rates with the 95% confidence intervals to estimate the effect of the pandemic.

## BMI, HbA1c, and liver function assessments

In both periods of the pandemic, we observed reductions in numbers of people receiving BMI, HbA1c, and liver function assessments before they were diagnosed with pancreatic cancer (*Figure 3*). For BMI, these were 20% and 14% less people than predicted received at least one assessment in the lockdown and recovery periods, respectively. For HbA1c, these were 10% and 8% reductions and for liver function these were 4% and 3% reductions, respectively (*Table 2*). Except for BMI, we did not observe reductions in numbers of people receiving these assessments after they were diagnosed with cancer.

## Symptoms and treatments

The reporting of jaundice in primary care and abdominal imaging in secondary care were reduced for people diagnosed in the lockdown period. For every 100 people diagnosed in the lockdown period, 3 (28%) less people (7 vs 10 [95% CI: 9–11] predicted) reported jaundice before they were diagnosed (*Table 2*). For abdominal imagining the observed in the lockdown period average reduction of 5% (29 vs 30 [95% CI: 27–33] predicted) did not reach statistical significance because it was the most pronounced only in the first 6 months of the lockdown period (*Figure 4B*). The reductions in both services were transient. They recovered to the pre-pandemic levels by April 2021 (jaundice) and by August 2020 (abdominal imaging) (*Figure 4A, B*).

*Figure 4C, D* represents two pancreatic cancer treatments. The prescribing of pancreatic enzyme replacement in primary care was not affected by the pandemic. However, pancreatic cancer resection (surgery) was significantly affected in both periods of the pandemic. For every 100 people diagnosed, 6 people were recorded as having received the resection within 6 months after pancreatic cancer diagnosis. We estimated that this was two people less than the predicted 8 (95% CI: 6–10) in the lockdown period and 9 (95% CI: 7–11) in the recovery period and represented an over 25% reduction in the number of people who received surgical resection during the pandemic as compared to what would be expected based on the pracademic trends.

## Emergency department visits, hospital admissions, and deaths

In the cohort of people diagnosed with pancreatic cancer, we found no effect of the COVID-19 pandemic on the number of emergency department visits and hospitalisations within 6 months before

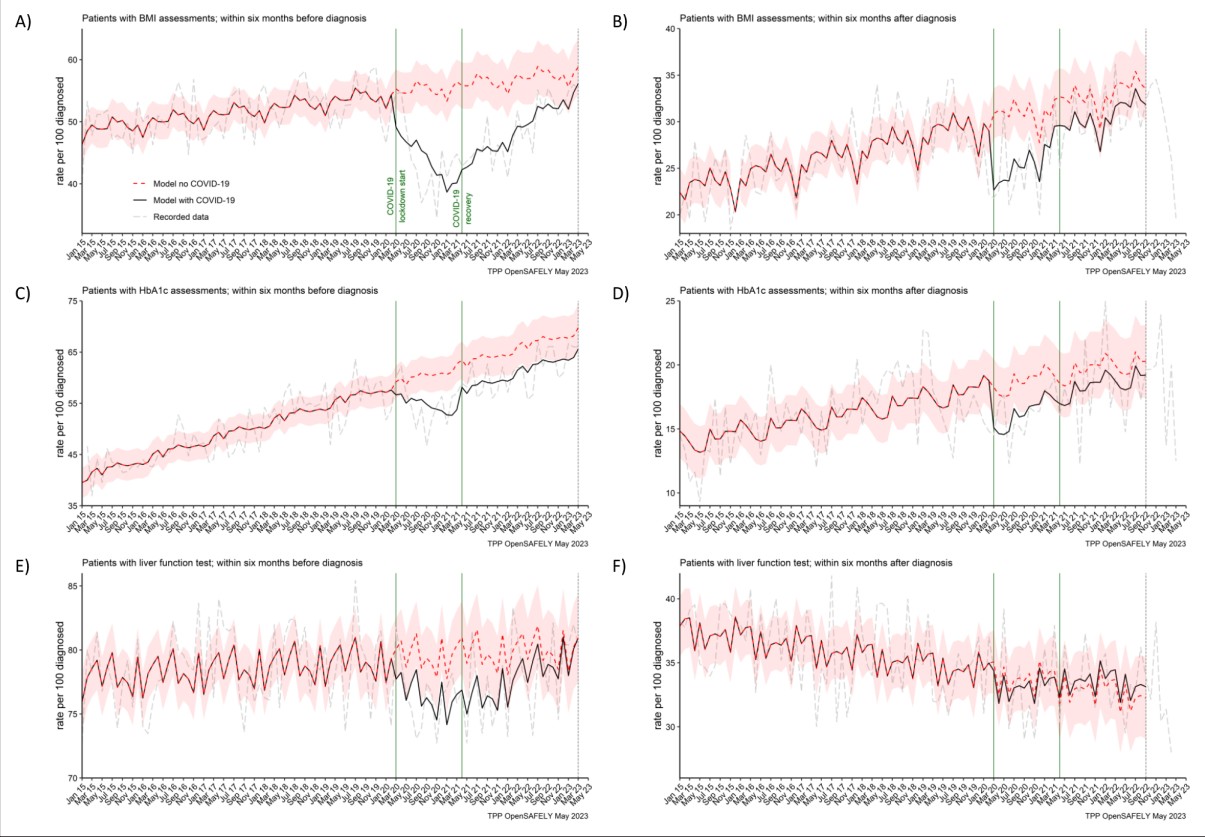

**Figure 3.** BMI, HbA1c and liver function testing decreased especially prior pancreatic cancer diagnosis. The observed and predicted (if the COVID-19 pandemic had not happened) monthly rates of (**A**) and (**B**) people with at least one body mass index (BMI) assessment before and after pancreatic cancer diagnosis, (**C**) and (**D**) people with at least one glycated haemoglobin (HbA1c) assessment before and after pancreatic cancer diagnosis, and (**E**) and (**F**) people with at least one liver function assessment before and after pancreatic cancer diagnosis. All rates are per 100 people diagnosed with pancreatic cancer. Generalised linear models were used to predict monthly rates with the 95% confidence intervals to estimate the effect of the pandemic.

or 6 months after pancreatic cancer diagnosis, and the numbers of recorded deaths within 6 months after pancreatic cancer diagnosis (*Figure 5*). The rates did not differ from what would be expected if the pandemic had not occurred (*Table 2*). The decrease in rates of deaths towards the end of the study period, visible from the graph, is most likely due to delays in deaths being entered into the registry data (*Office for National Statistics, 2021*).

## Discussion
### Summary and findings in context

We found that many of the pancreatic cancer-related services were disrupted across the pathway of care. This is in line with previous reports about healthcare being negatively affected by the COVID-19 pandemic (*Greenwood and Swanton, 2021*; *Patt et al., 2020*; *Richards et al., 2020*; *Morris et al., 2021*; *McKay et al., 2021*; *Glasbey et al., 2021*; *Nepogodiev et al., 2022*; *Sud et al., 2020*; *Earnshaw et al., 2020*; *Geh et al., 2022*; *Popovic et al., 2022*). Healthcare assessments, such as BMI, HbA1c, and liver function, were delivered to fewer people than would be expected if the pandemic had not occurred. This could impair not only the early diagnosis of pancreatic cancer, but also diagnosis of other diseases such as diabetes (*Lemanska et al., 2022*). In addition, this could have implications for the quality of routine data for research (*Lemanska et al., 2022*; *Garies et al., 2021*; *Staff et al., 2016*).

Disappointingly, we observed that by March 2023, 3 years into the pandemic, the pre-diagnosis testing of BMI and HbA1c did not recover to the pre-pandemic levels. However, reassuringly, for

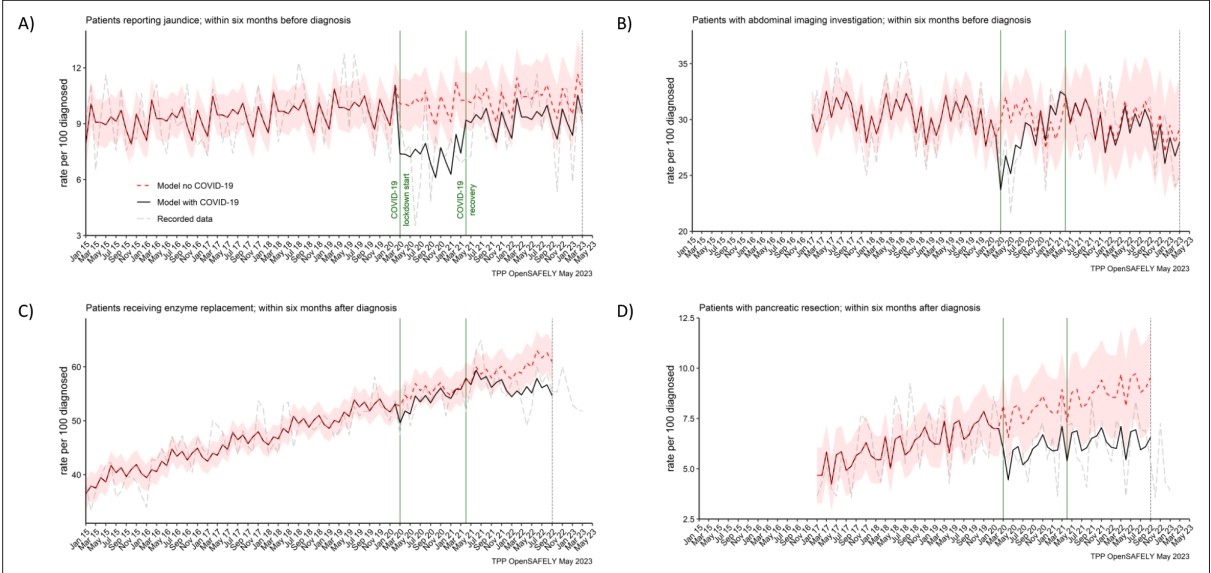

**Figure 4.** Recording of jaundice, abdominal imaging and surgical resections were negatively affected from the start of the pandemic, with surgical resections not recovering by the end of the study period. The observed and predicted (as if the COVID-19 pandemic had not happened) monthly rates of people (**A**) reporting jaundice before the diagnosis, (**B**) receiving abdominal imagining assessment before the diagnosis, (**C**) receiving pancreatic enzyme supplementation after the diagnosis, and (**D**) pancreatic resection within after the diagnosis. Generalised linear models were used to predict monthly rates with the 95% confidence intervals to estimate the effect of the pandemic.

people who received pancreatic cancer diagnosis (post-diagnosis testing of BMI, HbA1c, and liver function), the primary care healthcare in this area was much more resilient. We observed that the effect of the pandemic on these tests for people already diagnosed was more transient, and after an initial drop in BMI and HbA1c at the start of the pandemic, the levels recovered by the end of the lockdown period (March 2021). More reassuringly also, the other services that after an initial decrease recovered by the end of the lockdown period, included consultations for jaundice in primary care and abdominal imaging in secondary care. It has been previously reported that many non-emergency diagnostic services, such as abdominal imaging, were suspended during the first peak of the pandemic, but they gradually reopened starting from July 2020 following publications of infection control guidelines (*Rees et al., 2020*).

The number of people recorded as diagnosed with pancreatic cancer was not affected. This is a positive finding but sets pancreatic cancer apart from the other major cancer sites such as breast (*Patt et al., 2020*; *Richards et al., 2020*), prostate (*Patt et al., 2020*; *Vardhanabhuti and Ng, 2021*), or colorectal (*Patt et al., 2020*; *Richards et al., 2020*; *Morris et al., 2021*; *Vardhanabhuti and Ng, 2021*; *Mazidimoradi et al., 2023*). This may be because pancreatic cancer does not rely on screening programs and diagnostic services in primary care which were severely affected during the pandemic. With the emergency presentations remaining the main route of diagnosis for pancreatic cancer (*National Cancer Intelligence Network, 2013*), it is possible that these were less affected. However, because staging information was not available, our study was not equipped to evaluate the full effect of the COVID-19 pandemic on pancreatic cancer diagnosis. In a relatively small study, *Hall et al., 2023* demonstrated that nearly a quarter of people less in the pandemic than before the pandemic was recommended for surgery. This could be because they were diagnosed with a more advanced cancer. Therefore, to better understand the effect of the COVID-19 pandemic on pancreatic cancer diagnosis, more research is needed to assess the staging information.

Similarly, although we showed that the number of deaths within 6 months after pancreatic cancer diagnosis was not affected, the conclusions that could be made about the length of survival were limited. *Madge et al., 2022* showed the length of survival reducing by more than a half from 7.4 months before the pandemic to 3.3 months during the pandemic. *Hall et al., 2023* demonstrated no difference in survival and reported 3.5 months survival for people diagnosed between March 2020 and May 2020 versus 4.4 months for people diagnosed in January 2019 to March 2019. Therefore,

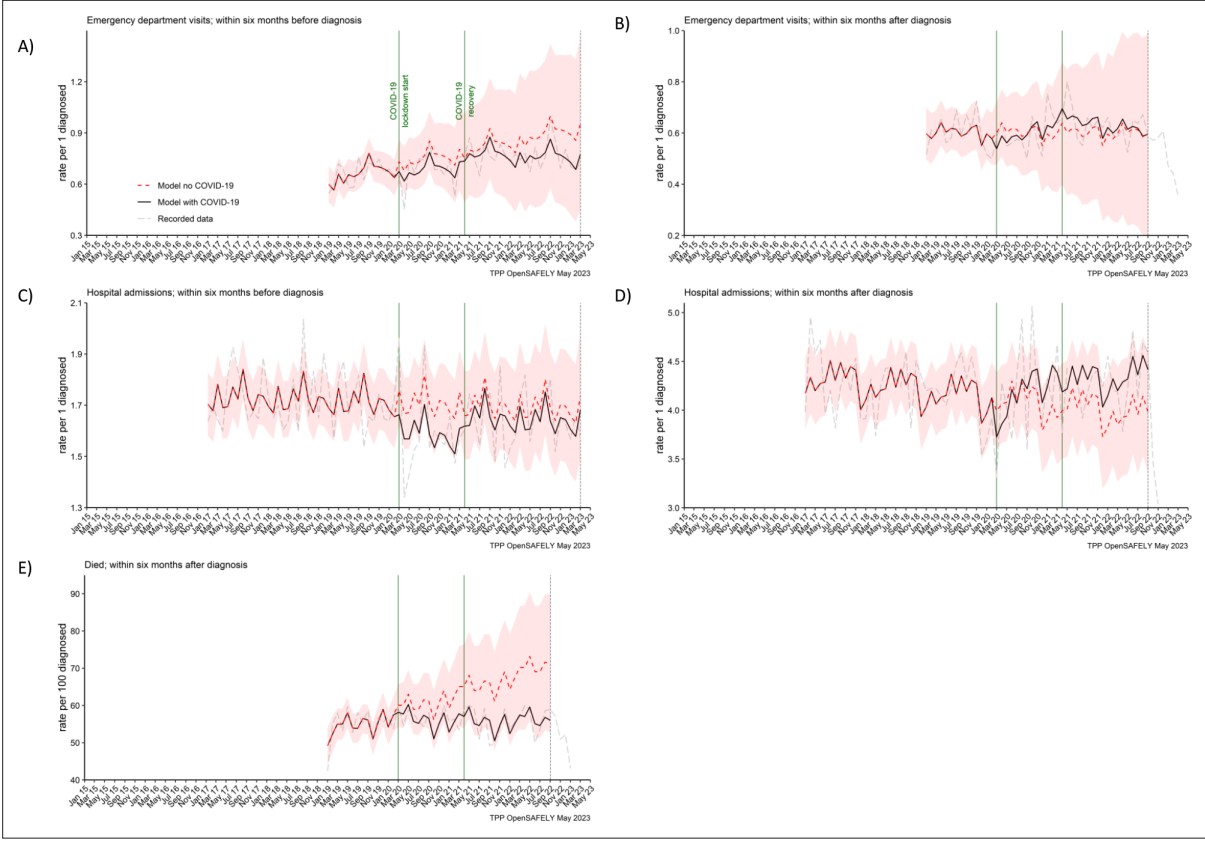

**Figure 5.** Emergency department visits, hospital admissions and deaths were not affected for people with pancreatic cancer. The observed and predicted (as if the COVID-19 pandemic had not happened) monthly rates of episodes per 1 participant diagnosed with pancreatic cancer, (**A**) emergency department visits within 6 months before the diagnosis, (**B**) emergency department visits within 6 months after, (**C**) hospital admissions within 6 months before, (**D**) hospital admissions within 6 months after, and (**E**) deaths within 6 months after per 100 people diagnosed with pancreatic cancer. Generalised linear models were used to predict monthly rates with the 95% confidence intervals to estimate the effect of the pandemic.

more research is needed to assess the effect of the pandemic on the length of survival post-diagnosis. In addition, the ongoing and future effects of the COVID-19 pandemic on key outcomes of pancreatic cancer such as the stage at diagnosis and the length of survival are still to be elucidated.

Additionally, contrary to the evidence from other patient groups, in this cohort, we did not find an effect of the COVID-19 pandemic on diabetes diagnosis. While in the general population, there was a significant reduction in diabetes diagnosis and management services (*Carr et al., 2021*).

Most people (over 90%) were in contact with primary care before and after pancreatic cancer diagnosis, and this was not affected by the pandemic. However, as compared to the pre-pandemic levels, we observed that the number of contacts in the pandemic increased on average by 1–2 contacts per person. This may reflect a true increase of contacts as primary care transitioned to remote consulting with an increased role of telephone triage (*Greenhalgh and Rosen, 2021*). However, this may also reflect increased recording of contacts in electronic health records as practices responded to the pandemic using different online systems with automated code recording activity (*Fisher, 2022*). Remote consulting offers an important advantage for improved efficiency and access not only during pandemics (*Mold et al., 2021*). However, it has been shown that for complex conditions, remote consulting can be a more time-consuming approach, increasing workload with subsequent follow-up appointments (*Salisbury et al., 2020*; *Murphy et al., 2021*).

## Strengths and limitations

The OpenSAFELY-TPP dataset is a population-based and nationally representative dataset of an unprecedented size and completeness. It also offers access to primary care records linked with hospital and mortality data. Therefore, the OpenSAFELY-TPP dataset was the key strength of this

study (*Andrews et al., 2022*). Also the cohort study design was a strength because we were able to focus specifically on people affected by pancreatic cancer, rather than investigate the already well known effect of COVID-19 on general practice (*The Health Foundation, 2022*). The interrupted time series approach and modelling COVID-19 rates has an advantage over simply comparing COVID-19 rates to pre-COVID rates. This is because we could account for long-term trends and seasonal variation in healthcare data. We used data dating back to 2015 to model the trends which made the prediction of expected rates more accurate. This provided a less-biased estimate of the effect of the pandemic. The near real-time data available via OpenSAFELY enabled us to investigate the most recent trends and recovery from the pandemic. In addition, the automated audit of healthcare services that we have developed within the OpenSAFELY, can enable regular updates. All analytics software and code lists are shared openly and are available for inspection and reuse, providing opportunity for reproduction of this report and reducing duplicative efforts.

We also note some limitations. Lists of clinical codes used to extract data may not be exhaustive and may miss episodes of care. This could create a source of bias. To minimise this bias, we have ensured that in an iterative process of curation and checking for completeness, at least two researchers with clinical expertise took part in collating code lists. To ensure the right concepts were captured, we consulted with clinicians who have the specific expertise in the field. We applied the principles of open, transparent, and reproducible research and all our code lists are available for scrutiny and reuse via our public GitHub repository (https://github.com/opensafely/Pancreatic_cancer; *Lemanska, 2023*). Pancreatic cancer case ascertainment was via coding in primary care, rather than via linkage with cancer registry (the gold standard data source for cancer diagnoses). It is possible that with this approach some pancreatic cancer cases could have been missed or miscoded. However, in the UK, the information about cancer diagnosis is sent to primary care within the hospital discharge letters and therefore primary care is a valid source of these data (*Margulis et al., 2018*). The incidence rates in this study aligned with the published rates validating good ascertainment of cases.

## Policy implications and future research

Considering the worsening healthcare crisis, it is important to continue monitoring services to ensure the quality of healthcare and recovery from the pandemic. As we innovate and adapt healthcare, with infection control measures, digital health approaches and increasing remote consulting, it is important to evaluate the impact of these on cancer-related healthcare. In addition, more research is needed to investigate how these changes affected different groups of patients. There is evidence that the COVID-19 pandemic exacerbated healthcare inequalities (*Popovic et al., 2022*) and future work should include stratified analysis investigating different socio-demographic groups.

## Conclusions

The COVID-19 pandemic was an unprecedented global event, adding pressures to already overburdened healthcare systems, further exacerbating healthcare crises. Positive lessons could be learnt from the resilient healthcare services which continued to deliver healthcare undisrupted, or those initially affected, where active measures to recover the capacity and volume of care were implemented quickly and safely. On the other hand, the reductions in healthcare experienced by people with non-COVID-19 illnesses such as pancreatic cancer, bolster the argument that efforts should focus on addressing the unmet needs of people with cancer.

## Information governance and ethical approval

NHS England is the data controller for OpenSAFELY-TPP; TPP is the data processor; all study authors using OpenSAFELY have the approval of NHS England. This implementation of OpenSAFELY is hosted within the TPP environment which is accredited to the ISO 27001 information security standard and is NHS IG Toolkit compliant (*NHS Digital, 2020b*).

Patient data have been pseudonymised for analysis and linkage using industry standard cryptographic hashing techniques; all pseudonymised datasets transmitted for linkage onto OpenSAFELY are encrypted; access to the platform is via a virtual private network (VPN) connection, restricted to a small group of researchers; the researchers hold contracts with NHS England and only access the platform to initiate database queries and statistical models; all database activity is logged; only aggregate

statistical outputs leave the platform environment following best practice for anonymisation of results such as statistical disclosure control for low cell counts (*NHS Digital, 2020a*).

The OpenSAFELY research platform adheres to the obligations of the UK General Data Protection Regulation (GDPR) and the Data Protection Act 2018. In March 2020, the Secretary of State for Health and Social Care used powers under the UK Health Service (Control of Patient Information) Regulations 2002 (COPI) to require organisations to process confidential patient information for the purposes of protecting public health, providing healthcare services to the public and monitoring and managing the COVID-19 outbreak and incidents of exposure; this sets aside the requirement for patient consent (*GOV.UK, 2020*). This was extended in November 2022 for the NHS England OpenSAFELY COVID-19 research platform (*GOV.UK, 2022*). In some cases of data sharing, the common law duty of confidence is met using, for example, patient consent or support from the Health Research Authority (*Confidentiality Advisory Group, 2013*).

Taken together, these provide the legal bases to link patient datasets on the OpenSAFELY platform. GP practices, from which the primary care data are obtained, are required to share relevant health information to support the public health response to the pandemic, and have been informed of the OpenSAFELY analytics platform.

The study was approved by the Health Research Authority (Research Ethics Committee reference 20/LO/0651) and the London School of Hygiene and Tropical Medicine (London, UK) Ethics Board (reference 21863).

## Data access and verification

Access to the underlying identifiable and potentially re-identifiable pseudonymised electronic health record data is tightly governed by various legislative and regulatory frameworks and restricted by best practice. The data in OpenSAFELY are drawn from General Practice data across England where TPP is the data processor. TPP developers initiate an automated process to create pseudonymised records in the core OpenSAFELY database, which are copies of key structured data tables in the identifiable records. These pseudonymised records are linked onto key external data resources that have also been pseudonymised via SHA-512 one-way hashing of NHS numbers using a shared salt. Bennett Institute for Applied Data Science developers and PIs holding contracts with NHS England have access to the OpenSAFELY pseudonymised data tables as needed to develop the OpenSAFELY tools. These tools in turn enable researchers with OpenSAFELY data access agreements to write and execute code for data management and data analysis without direct access to the underlying raw pseudonymised patient data, and to review the outputs of this code. All code for the full data management pipeline—from raw data to completed results for this analysis is available at from https://github.com/opensafely/Pancreatic_cancer (*Lemanska, 2023*). The data management and analysis code for this paper were led by AL and contributed to by CA.

## Acknowledgements

We are very grateful for all the support received from the TPP Technical Operations team throughout this work, and for generous assistance from the information governance and database teams at NHS England and the NHS England Transformation Directorate. Membership of the OpenSAFELY Collaborative: Alex J Walker, Brian MacKenna, Peter Inglesby, Christopher T Rentsch, Helen J Curtis, Caroline E Morton, Jessica Morley, Amir Mehrkar, Seb Bacon, George Hickman, Chris Bates, Richard Croker, David Evans, Tom Ward, Jonathan Cockburn, Simon Davy, Krishnan Bhaskaran, Anna Schultze, Elizabeth J Williamson, William J Hulme, Helen I McDonald, Laurie Tomlinson, Rohini Mathur, Rosalind M Eggo, Kevin Wing, Angel YS Wong, Harriet Forbes, John Tazare, John Parry, Frank Hester, Sam Harper, Ian J Douglas, Stephen JW Evans, Liam Smeeth, and Ben Goldacre. Funding statement This work was supported by the Wellcome Trust grant number 222097/Z/20/Z; Medical Research Council (MRC) grant numbers MR/V015757/1, MC_PC-20059, MR/W016729/1; National Institute for Health and Care Research (NIHR) grant numbers NIHR135559, COV-LT2-0073, and Health Data Research UK grant numbers HDRUK2021.000, HDRUK2021.0157. This work was also supported by the MRC grant number MR/W021390/1 as part of the postdoctoral fellowship awarded to AL and undertaken at the Bennett Institute, University of Oxford. The views expressed are those of the authors and not necessarily those of the NIHR, NHS England, UK Health Security Agency (UKHSA), or the Department

of Health and Social Care. Funders had no role in the study design, collection, analysis, and interpretation of data; in the writing of the report; and in the decision to submit the article for publication.

## Additional information

### Group author details

**The OpenSAFELY Collaborative**
Alex J Walker; Brian MacKenna; Peter Inglesby; Christopher T Rentsch; Helen J Curtis; Caroline E Morton; Jessica Morley; Amir Mehrkar; Seb Bacon; George Hickman; Chris Bates; Richard Croker; David Evans; Tom Ward; Jonathan Cockburn; Simon Davy; Krishnan Bhaskaran; Anna Schultze; Elizabeth J Williamson; William J Hulme; Helen I McDonald; Laurie Tomlinson; Rohini Mathur; Rosalind M Eggo; Kevin Wing; Angel YS Wong; Harriet Forbes; John Tazare; John Parry; Frank Hester; Sam Harper; Ian J Douglas; Stephen JW Evans; Liam Smeeth; Ben Goldacre

### Competing interests

Ben Goldacre: received research funding from the Laura and John Arnold Foundation, the NHS National Institute for Health Research (NIHR), the NIHR School of Primary Care Research, NHS England, the NIHR Oxford Biomedical Research Centre, the Mohn-Westlake Foundation, NIHR Applied Research Collaboration Oxford and Thames Valley, the Wellcome Trust, the Good Thinking Foundation, Health Data Research UK, the Health Foundation, the World Health Organisation, UKRI MRC, Asthma UK, the British Lung Foundation, and the Longitudinal Health and Wellbeing strand of the National Core Studies programme; he is a Non-Executive Director at NHS Digital; he also receives personal income from speaking and writing for lay audiences on the misuse of science. Brian MacKenna: is employed as a pharmacist by NHS England and seconded to the Bennett Institute. The author is a trustee of a charity ICAP. The authors has no other competing interests to declare. The OpenSAFELY Collaborative: The other authors declare that no competing interests exist.

### Funding

| Funder | Grant reference number | Author |
|---|---|---|
| Medical Research Council | MR/W021390/1 | Agnieszka Lemanska |
| Wellcome Trust | 222097/Z/20/Z | Ben Goldacre |
| Medical Research Council | MR/V015757/1 | Ben Goldacre |
| National Institute for Health and Care Research | NIHR135559 | Ben Goldacre |
| Health Data Research UK | HDRUK2021.000 | Ben Goldacre |
| Medical Research Council | MC_PC-20059 | Ben Goldacre |
| Medical Research Council | MR/W016729/1 | Ben Goldacre |
| National Institute for Health and Care Research | COV-LT2-0073 | Ben Goldacre |
| Health Data Research UK | 2021.0157 | Ben Goldacre |

The funders had no role in study design, data collection and interpretation, or the decision to submit the work for publication. For the purpose of Open Access, the authors have applied a CC BY public copyright license to any Author Accepted Manuscript version arising from this submission.

### Author contributions

Agnieszka Lemanska, Conceptualization, Data curation, Software, Formal analysis, Funding acquisition, Investigation, Visualization, Methodology, Writing – original draft, Writing – review and editing; Colm Andrews, Data curation, Software, Formal analysis, Methodology, Writing – original draft, Project administration; Louis Fisher, Data curation, Software, Formal analysis; Seb Bacon, Peter Inglesby, Simon Davy, Software; Adam E Frampton, Supervision, Validation, Investigation, Writing

– review and editing; Amir Mehrkar, Conceptualization, Resources, Investigation, Project adminis-
tration; Keith Roberts, Supervision, Validation, Visualization, Writing – review and editing; Praveetha
Patalay, Conceptualization, Resources, Supervision, Funding acquisition, Writing – review and editing;
Ben Goldacre, Conceptualization, Resources, Supervision, Funding acquisition, Investigation, Meth-
odology, Project administration; Brian MacKenna, Conceptualization, Resources, Data curation,
Formal analysis, Supervision, Funding acquisition, Investigation, Methodology, Writing – original draft,
Project administration, Writing – review and editing; The OpenSAFELY Collaborative, Conceptualiza-
tion, Data curation, Funding acquisition, Methodology, Project administration, Resources, Software,
Validation; Alex J Walker, Conceptualization, Resources, Data curation, Supervision, Funding acqui-
sition, Investigation, Methodology, Writing – original draft, Project administration, Writing – review
and editing

## Author ORCIDs
Agnieszka Lemanska (ID) https://orcid.org/0000-0003-4849-2430

## Ethics
The study was approved by the Health Research Authority (Research Ethics Committee reference 20/
LO/0651) and the London School of Hygiene and Tropical Medicine (London, UK) Ethics Board (refer-
ence 21863).

## Decision letter and Author response
Decision letter https://doi.org/10.7554/eLife.85332.sa1
Author response https://doi.org/10.7554/eLife.85332.sa2

---

## Additional files

### Supplementary files
• MDAR checklist
• Supplementary file 1. Study flowchart diagram.

### Data availability
Data used in this project are pseudonymised patient level data that are potentially re-identifiable and
therefore cannot be shared or deposited open access. Data are available via the OpenSAFELY plat-
form with the approval of NHS England who is the data controller. Researchers applying for access
to data for their projects are requested to submit a protocol and complete a data request form (as
described here: https://www.opensafely.org/onboarding-new-users/). The process is outlined in detail
in a flowchart https://www.opensafely.org/governance/os-workflow.jpg. Software for data extraction
and analysis, and code lists used to define variables, are shared openly for review and re-use at https://
github.com/opensafely/Pancreatic_cancer (copy archived at *Lemanska, 2023*).

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
