## [Editor Report]

This study provides useful information on the impact of the pandemic on the quantity of healthcare delivered to patients with pancreatic cancer in England. The authors showed that there was no difference in the number of diagnoses of pancreatic cancer during the pandemic compared to the preceding 5-year period, but a reduction in surgical resections by nearly 25%. They reported no difference in deaths between the two periods. They show no differences in rates of diagnosis, but the clinical relevance is incomplete as they have not compared survival from cancer between those time periods.

---

## [Decision Letter]

**Decision letter after peer review:**

Thank you for submitting your article "Healthcare in England was affected by the COVID-19 pandemic across the pancreatic cancer pathway: a cohort study using OpenSAFELY-TPP" for consideration by *eLife*. Your article has been reviewed by one peer reviewer, and I oversaw the evaluation in my dual role of Reviewing Editor and Senior Editor. The reviewer has opted to remain anonymous.

What follows below is an edited compilation of the essential and ancillary points provided by the reviewer. Please submit a revised version that addresses these concerns directly. Although we expect that you will address these comments in your response letter, we also need to see the corresponding revision clearly marked in the text of the manuscript. Some of the reviewer's comments may seem to be simple queries or challenges that do not prompt revisions to the text. Please keep in mind, however, that readers may have the same perspective as the reviewer. Therefore, it is essential that you amend or expand the text to clarify the narrative accordingly.

*Reviewer #1 (Recommendations for the authors):*

1. Why was the TPP dataset used, and how have the authors ensured that this does not introduce a degree of systematic bias given only 40% of the population in England are covered by this dataset?

2. What quality control was employed to ensure that the coding within the TPP dataset accurately reflected the exact diagnosis and stage of disease? This is listed as a limitation – could the authors specify how much of a limitation this was, and whether any attempt was made to assess the accuracy of coding, given this is a key element of the study?

3. One very important limitation of the study is that the true effects of the pandemic on pancreatic cancer outcomes cannot be assessed with the methodology of this study. Did the authors attempt to assess stage of pancreatic cancer at the time of diagnosis in those two time periods, or specifics of treatment? Was the reason for lack of surgical intervention in the 25% of patients given – if this was related to delayed diagnosis causing inoperability, this would be a significant finding and would render the conclusion that pancreatic cancer diagnosis was not affected somewhat irrelevant in the wider clinical context, if patients were ultimately less likely to survive the disease if diagnosed during the pandemic.

4. Related to the above, the authors should reference other studies on outcomes from pancreatic cancer during the pandemic and reflect on their findings in this context.

---

## [Author Response]

Reviewer #1 (Recommendations for the authors):1. Why was the TPP dataset used, and how have the authors ensured that this does not introduce a degree of systematic bias given only 40% of the population in England are covered by this dataset?

Methods Section. We added:

“We used the nationally representative OpenSAFELY-TPP dataset comprising 24 million people currently registered with primary care practices that use TPP’s SystmOne software (covering over 40% of England’s population). This dataset was used for this project because of its unprecedented size, because it is nationally representative [32], and because it enables access to primary care records linked to hospital records and mortality data.”

Strengths and Limitation Section, we added:

“The OpenSAFELY-TPP dataset is a population-based and nationally representative dataset of an unprecedented size and completeness. It also offers access to primary care records linked with hospital and mortality data. Therefore, the OpenSAFELY-TPP dataset was the key strength of this study [32].”

2. What quality control was employed to ensure that the coding within the TPP dataset accurately reflected the exact diagnosis and stage of disease? This is listed as a limitation – could the authors specify how much of a limitation this was, and whether any attempt was made to assess the accuracy of coding, given this is a key element of the study?

Strengths and Limitation Section, we added:

“Lists of clinical codes used to extract data may not be exhaustive and may miss episodes of care. This could create a source of bias. To minimise this bias, we have ensured that in an iterative process of curation and checking for completeness, at least two researchers with clinical expertise took part in collating code lists. To ensure the right concepts were captured, we consulted with clinicians who have the specific expertise in the field. We applied the principles of open, transparent, and reproducible research and all our code lists are available for scrutiny and reuse via our public GitHub repository (github.com/opensafely/Pancreatic_cancer).”

3. One very important limitation of the study is that the true effects of the pandemic on pancreatic cancer outcomes cannot be assessed with the methodology of this study. Did the authors attempt to assess stage of pancreatic cancer at the time of diagnosis in those two time periods, or specifics of treatment? Was the reason for lack of surgical intervention in the 25% of patients given – if this was related to delayed diagnosis causing inoperability, this would be a significant finding and would render the conclusion that pancreatic cancer diagnosis was not affected somewhat irrelevant in the wider clinical context, if patients were ultimately less likely to survive the disease if diagnosed during the pandemic.

Results section, we amended:

We did not observe an effect of the COVID-19 pandemic on pancreatic cancer diagnosis (Figure 1A).

To

“We did not observe an effect of the COVID-19 pandemic on the number of people recorded as diagnosed with pancreatic cancer (Figure 1A).”

Discussion section, we amended:

Pancreatic cancer diagnosis was not affected.

To

“The number of people recorded as diagnosed with pancreatic cancer was not affected.”

Discussion section, we added:

“However, because staging information was not available, our study was not equipped to evaluate the full effect of the COVID-19 pandemic on pancreatic cancer diagnosis. In a relatively small study, Hall et al. 2023 [39] demonstrated that nearly a quarter of people less in the pandemic than before the pandemic was recommended for surgery. This could be because they were diagnosed with a more advanced cancer. Therefore, to better understand the effect of the COVID-19 pandemic on pancreatic cancer diagnosis, more research is needed to assess the staging information.

Similarly, although we showed that the number of deaths within six months after pancreatic cancer diagnosis was not affected, the conclusions that could be made about the length of survival were limited. Madge et al. (2022) [40] showed the length of survival reducing by more than a half from 7.4 months before the pandemic to 3.3 months during the pandemic. While Hall et al. 2023 [39] demonstrated no difference in survival and reported 3.5 months survival for people diagnosed between March 2020 and May 2020 versus 4.4 months for people diagnosed in January 2019 to March 2019. Therefore, more research is needed to assess the effect of the pandemic on the length of survival post diagnosis. In addition, the ongoing and future effects of the COVID-19 pandemic on key outcomes of pancreatic cancer such as the stage at diagnosis and the length of survival are still to be elucidated.”

4. Related to the above, the authors should reference other studies on outcomes from pancreatic cancer during the pandemic and reflect on their findings in this context.

The following references were added to support the discussion points on diagnosis and survival:

Madge, O., et al., The COVID-19 Pandemic Is Associated with Reduced Survival after Pancreatic Ductal Adenocarcinoma Diagnosis: A Single-Centre Retrospective Analysis. J Clin Med, 2022. 11(9).

Hall, L.A., et al., The impact of the COVID-19 pandemic upon pancreatic cancer treatment (CONTACT Study): a UK national observational cohort study. British Journal of Cancer, 2023. 128(10): p. 1922-1932.